# Oral *Candida* spp. Colonisation Is a Risk Factor for Severe Oral Mucositis in Patients Undergoing Radiotherapy for Head & Neck Cancer: Results from a Multidisciplinary Mono-Institutional Prospective Observational Study

**DOI:** 10.3390/cancers14194746

**Published:** 2022-09-29

**Authors:** Cosimo Rupe, Gioele Gioco, Giovanni Almadori, Jacopo Galli, Francesco Micciché, Michela Olivieri, Massimo Cordaro, Carlo Lajolo

**Affiliations:** 1Head and Neck Department, Fondazione Policlinico Universitario A. Gemelli—IRCCS, School of Dentistry, Università Cattolica del Sacro Cuore, Largo A. Gemelli, 8, 00168 Rome, Italy; 2Head and Neck Department, Fondazione Policlinico Universitario A. Gemelli—IRCCS, Institute of Otolaryngology, Università Cattolica del Sacro Cuore, Largo A. Gemelli, 8, 00168 Rome, Italy; 3Department of Radiation Oncology, Fondazione Policlinico Universitario A. Gemelli—IRCCS, Institute of Radiology, Università Cattolica del Sacro Cuore, Largo A. Gemelli, 8, 00168 Rome, Italy

**Keywords:** oral mucositis, radiotherapy, head and neck cancer, oral *Candida* spp., oral candidiasis, chemotherapy, radiochemotherapy

## Abstract

**Simple Summary:**

This study aims to find a correlation between *Candida* spp. oral colonisation prior to radiotherapy and (i) the development of severe oral mucositis (OM) (grade 3/4) and (ii) early development of severe OM (EOM). *Candida* spp. in the oral cavity appears to be a predictive factor of EOM. Preventive treatment could aid in reducing incidence of EOM. Further clinical trials are required to confirm our findings.

**Abstract:**

Background: This study aims to find a correlation between *Candida* spp. oral colonisation prior to radiotherapy (RT) and (i) the development of severe oral mucositis (OM) (grade 3/4) and (ii) early development of severe OM (EOM). Methods: The protocol was registered on ClinicalTrials.gov (ID: NCT04009161) and approved by the ethical committee of the ‘Fondazione Policlinico Universitario Gemelli IRCCS’ (22858/18). An oral swab was obtained before RT to assess the presence of *Candida* spp. Severe OM occurring before a dose of 40 Gy was defined as EOM. Results: No patient developed G4 OM, and only 36/152 patients (23.7%) developed G3 OM. Tumour site and lymphocytopenia were risk factors for severe OM (OR for tumour site: 1.29, 95% CI: 1–1.67, *p* = 0.05; OR for lymphocytopenia: 8.2, 95% CI: 1.2–55.8, *p* = 0.03). We found a correlation between *Candida* spp. and EOM (OR: 5.13; 95% CI: 1.23–21.4 *p* = 0.04). Patients with oral colonisation of *Candida* spp. developed severe OM at a mean dose of 38.3 Gy (range: 28–58; SD: 7.6), while negative patients did so at a mean dose of 45.6 Gy (range: 30–66; SD: 11.1). Conclusions: *Candida* spp. in the oral cavity appears to be a predictive factor of EOM.

## 1. Introduction

More than 900,000 new cases of head–neck cancer (HNC) are diagnosed worldwide, with 40,000 new cases and 7890 deaths reported annually in the United States. HNC can arise in multiple anatomic subsites (i.e., the oral cavity, oropharynx, hypopharynx, nasopharynx, larynx, and salivary glands) [1].

HNC treatment is challenging and requires a multidisciplinary approach with a team of specialists, including head and neck surgeons, radiation oncologists, medical oncologists, nutritionists, nuclear physicians, and oral oncologists [2,3].

Approximately 60% of patients with HNC require radiotherapy (RT), with or without induction chemotherapy [4], and a substantial proportion of patients suffer significant treatment-related adverse effects [5], including acute adverse effects (i.e., mucositis and dermatitis) that occur during treatment and late adverse effects (i.e., dysgeusia, osteoradionecrosis, and trismus) that occur in the weeks following the end of therapy [6,7,8]. 

Oral and oropharyngeal mucositis (OM) caused by RT and combined systemic therapies appears to be a significant side effect that presents numerous clinical signs and symptoms [9]. It affects the patient’s quality of life (QoL) and is associated with symptoms such as pain, bleeding, dysphagia, local infections, increased susceptibility to secondary and systemic infections, impaired food intake, and weight loss [10,11]. The incidence of OM in patients treated with RT is estimated to be approximately 80%, becoming nearly ubiquitous in patients undergoing radiochemotherapy (RTCT) [12]. 

Severe OM (grade 3/4), according to the Radiation Therapy Oncology Group (RTOG), appears in approximately 43% of patients undergoing combination treatment [13]. It may cause inadequate food intake; this further results in the development of severe nutritional deficiencies and a need for parenteral nutrition. In addition, approximately 15% of patients require an interruption of RT or dose reshaping of concomitant systemic therapy, thus influencing the effectiveness of the treatment [14]. However, the incidence and severity of this condition varies depending on several factors (i.e., cancer subsite, radiation dose, volume of the irradiated mucosa, daily fractionation, association with CT, habit history, oral health prior to initiation of treatment, and neutrophil recovery period) [15].

Nevertheless, while dosimetric parameters are best known to correlate with the time of onset and severity of side effects, the available literature does not provide clear evidence about the clinical parameters that can predict OM development or may indicate worsening of OM [16].

Oral candidiasis is a common fungal disease caused by overgrowth of *Candida* spp. in the mouth. Acute pseudomembranous candidiasis and acute erythematous candidiasis are the most frequent clinical patterns of oral candidiasis, requiring complex treatments (i.e., adequate oral hygiene, topical agents, and systemic medications) that often lead to chronic candidiasis in patients with HNC [17,18]. Oral candidiasis, especially in its acute-erythematous manifestation, may enhance OM-related symptoms and result in worsening of the clinical condition of patients. Thus, treatment of oral candidiasis is recommended when RT in the head and neck region is scheduled. Nevertheless, *Candida* spp. can be found in 50% of the population as a component of the oral microbiota, and it can become a pathogen even after the initiation of RT [19]. Furthermore, alterations in the mucosal layer structure caused by OM often allow bacteria and fungi to penetrate damaged tissue and cause infections, increasing the risk of oral candidiasis development [20].

The primary objective of this observational prospective cohort study was to understand whether the presence of *Candida* spp. in the oral cavity, evaluated using an oral swab taken prior to initiation of RT, is a risk factor for the development of severe OM during RT. The secondary objectives were (i) to understand whether oral colonisation of *Candida* spp. is a risk factor for the early development of severe OM (EOM), defined as an OM developed at 40 Gy of the cumulative radiation dose, (ii) to understand whether other clinical parameters (radiation dose, dose received by the oral cavity and oropharynx, smoking history, white blood cell count (WBC), chemotherapy (CT), and cancer subsite) are risk factors for the development of severe OM or (iii) EOM, and (iv) to evaluate the overall incidence of OM in the studied cohort.

## 2. Materials and Methods

### 2.1. Setting

The protocol was registered on ClinicalTrials.gov (ID: NCT04009161) and was approved by the ethical committee of the ‘Fondazione Policlinico Universitario A. Gemelli IRCCS in Rome’ (22858/18). The study was conducted in accordance with the Declaration of Helsinki, and all patients signed an informed consent form. Patients with HNC seeking treatment at the Oral Medicine, Head and Neck Department, with a scheduled external beam RT at Gemelli Advanced Radiation Therapy (ART), Fondazione Policlinico Universitario A. Gemelli-IRCSS, between March 2017 and August 2021, were consecutively recruited for this study. All included patients visited the hospital prior to initiation of RT. This paper was written in accordance with the STROBE guidelines (Appendix A).

### 2.2. Participants

The inclusion criteria were HNC diagnosis, indication for RT (either adjuvant or neoadjuvant), and treatment with a curative intent. The exclusion criteria were as follows: indication for palliative treatment, presence of clinically detectable signs of oral candidiasis, patients who received neoadjuvant CT before RT, and metastatic disease. 

### 2.3. Variables—Anamnesis

Before the clinical examination, anagraphic and anamnestic data (age, sex, and comorbidities) were recorded, particularly focusing on the oncologic history of the patient (tumour site, histological type of cancer, stage of the tumour, and previous oncologic treatments) and on the exposure to risk factors for the oncologic disease (i.e., smoking).

### 2.4. Variables—Oral Examination

Subsequently, clinical evaluation of the oral mucosal conditions was performed, focusing on the presence of signs of oral candidiasis. 

Furthermore, oral colonisation by *Candida* spp. was recorded using a sterile swab (eSwab^®®^, Copan’s Liquid Amies Elution Swab, Copan Italia SPA, Brescia, Italy); it was rubbed on the following mucosal surfaces of the oral cavity: hard palate, tongue, upper and lower vestibule, and ending at the commissures of the mouth. Post sample collection, sterile swabs were placed in tubes containing 1 mL of transport medium. The tubes were then stored at 4 °C until further processing. Processing involved streak inoculation of the swab onto Sabouraud dextrose agar (SDA) plates, followed by incubation at 37 °C for 48 h, according to the manufacturer’s instructions. 

The unstimulated salivary flow rate was assessed using the spitting method. Patients were instructed to collect saliva for 5 min in a graded tube. The stimulated salivary flow was determined in a similar manner. Saliva secretion was stimulated by applying a solution of 2% citric acid to the sides of the tongue at intervals of 30 s. An unstimulated salivary flow (USF) of over 0.4 mL/min was considered normal [21]. Furthermore, a blood count was performed before the beginning of RT, and the following variables were recorded: number of leukocytes, neutrophils, and lymphocytes. Leukopenia was defined as a leukocyte count < 4 × 10^9^/L, neutropenia as <1.5 × 10^9^/L neutrophils, and lymphocytopenia as <1 × 10^9^/L lymphocytes.

### 2.5. Variables—RT and OM

RT was delivered using the volumetric-modulated arc radiotherapy (VMAT) technique with a linear accelerator, and treatment was administered in five daily fractions per week for 6–7 weeks. The definition of volume is in accordance with international RT guidelines [22,23].

The treatment plan was optimised to ensure adequate coverage of the target (D95% of the treatment volume received > 95% of the prescribed dose) and to respect the constraints of the various organs at risk, identified during contouring.

Treatment included a daily image-guided radiation therapy (IGRT) check with Cone-Beam CT and, if necessary, the radiation dose was re-planned between 30 Gy and 40 Gy, in case of tumour shrinkage or anatomical changes. Patients were instructed to receive supportive therapy according to the centre’s procedures and international guidelines [24]. Each patient underwent at least one weekly examination during RT, in which the diagnosis of OM took place: if present, OM was recorded according to the National Cancer Institute Common Toxicity Criteria for Adverse Events (CTCae, version 4.0) [25]. Each patient was assigned a single OM grade, corresponding to the most severe OM grade recorded during RT and during the immediate RT follow-up. When OM signs disappeared, the patients terminated their study involvement. For patients who developed severe OM, the dose at which OM developed was also recorded.

According to Mallick et al., since the onset of G3-G4 OM occurs between 50 Gy and 60 Gy [26], we assumed that the onset of severe toxicity at 40 Gy should be considered as early acute toxicity. Onset of OM at a dose of 40 Gy or less was defined as an ‘early onset mucositis (EOM)’.

Severe OM was managed according to the centre’s procedures and international guidelines: in case of G3 OM, the patients were treated by analgesic drugs to reduce the pain [24].

### 2.6. Statistical Analysis

The sample size was calculated, with a 90% confidence level and 80% power, by comparing two proportions: considering 45% as the expected incidence of severe OM in the presence of *Candida* spp. in the oral cavity and 25% as the expected incidence in the absence of oral colonisation of *Candida* spp. The required sample size was 136 patients, with 68 in each group (positive or negative for oral cavity swabs). Considering a dropout rate of 10%, the final sample size was 150 patients.

The following variables were recorded as baseline patient characteristics (sex, age, histological type and stage of the tumour, site of the tumour, risk factors such as smoking history, previous oncological surgery, salivary flow, presence of *Candida* spp. oral colonisation), basal treatment characteristics (scheduled CT, Total Radiation Dose, daily fraction, dose received by the oral cavity and oropharynx), and treatment-related toxicity parameters (presence and grade of OM).

Qualitative variables were described using absolute and percentage frequencies, while the Kolmogorov–Smirnov test was performed to evaluate the normal distribution of quantitative variables. Quantitative variables were summarised either as mean and standard deviation (SD) if normally distributed, or as median and percentiles otherwise.

OM was reclassified into three categories: absence of OM, grade I or II OM, and grade 3 or 4 OM. 

Correlation analysis between OM onset and clinical characteristics of the patients was performed. The Mann–Whitney U test and Kruskall–Wallis test were performed to compare the continuous variables with non-parametric distribution, while the parametric variables were analysed through an ANOVA test; Pearson’s χ^2^ test and Fisher’s exact test were used to compare the discontinuous variables. Statistical analysis was stratified according to the following variables: development of severe OM and early onset of severe OM.

Univariate analysis was performed to determine the risk factors associated with the onset of OM, and risk factors were introduced in a stepwise logistic regression analysis to identify independent predictors of OM. The same statistical analysis was used to determine risk factors for EOM. All statistical analyses were performed using IBM SPSS Statistics software (IBM Corp. Released 2017. IBM SPSS Statistics for Apple, Version 25.0 (IBM Corp., Armonk, NY, USA).

## 3. Results

One hundred and sixty-three patients were enrolled in the study; 11 patients were excluded per the inclusion criteria: 5 patients suffered from oral candidiasis at baseline, so they were treated but excluded from the final sample, whereas 6 patients had received a planning for a palliative treatment. The final sample included 152 patients (49 female and 103 male), with a mean age of 60.3 years (range: 22–86). One hundred and fifteen (75.7%) patients had locally advanced oncologic disease (stage III–IV), 93 patients (61.2%) received treatment with curative intent, and 59 patients (38.8%) received adjuvant treatment. Oral cavity swabs were positive in 68 of the 152 patients (44.7%), and the remaining (84/152, 55.3%) swabs showed negative results prior to the initiation of RT. The mean total RT dose was 67.6 Gy (50–72), and the dose was fractionated in 2 Gy/die in majority of the patients (136/152, 89.5%). One hundred and twenty patients out of 152 (78.9% of the total sample) developed OM. Severe OM occurred in 36 patients (23.7% of the total sample). None of the patients developed G4 OM. Termination of RT before reaching the target dose was not required in any patient, and all patients completed their scheduled treatment; three patients had to discontinue RT for a few days. However, they still finished their RT course. Patient and treatment characteristics and related toxicity parameters are shown in Table 1.

The study flow chart is presented in Figure 1.

Results of the statistical analysis stratified according to the development of severe OM are shown in Table 2. 

In the univariate analysis, the clinical parameters associated with severe OM onset were the tumour site and RT-CT treatment. Patients with different tumour sites showed a different incidence of severe OM (χ^2^ test, *p* < 0.05), and nasopharyngeal cancers were associated with the highest incidence of OM (9/19 patients, 47.4%), while laryngeal cancers had the lowest incidence of OM (1/28, 3.6%). Patients who developed severe OM (25/36 patients; 69.4%) were more frequently treated with combined RT-CT (χ^2^ test, *p* = 0.05), while patients who did not develop severe OM often did not receive combined treatment (61/116; 52.6%). The prevalence of oral colonisation of *Candida* spp. was higher in patients with severe OM (20/36; 55.6%) than in other patients in the cohort (48/116; 41.4%), but the correlation was not statistically significant. Furthermore, severe OM correlated with leukopenia, neutropenia, and lymphocytopenia (χ^2^ test, *p* < 0.05). However, when inserted in a multiple logistic regression model, only tumour site and lymphocytopenia were statistically significant risk factors for severe OM development (OR for tumour site: 1.29, 95% CI: 1–1.67, *p* = 0.05; OR for lymphocytopenia: 8.2, 95% CI: 1.2–55.8, *p* = 0.03).

Table 3 summarises the characteristics of patients who developed EOM. Oral colonisation by *Candida* spp. was correlated with an early onset of severe OM in the univariate analysis (χ^2^ test, *p* < 0.05), showing that the presence of *Candida* spp. in the oral cavity is a risk factor for the development of EOM (OR: 5.13, 95% CI: 1.23–21.4 *p* = 0.04). Patients with oral colonisation by *Candida* spp. developed severe OM at a mean dose of 38.3 Gy (range: 28–58), while patients with negative oral swabs developed severe OM at a mean dose of 45.6 Gy (range: 30–66). From a clinical point of view, these patients experienced severe OM approximately four days before those with a negative swab.

## 4. Discussion

RT has played a fundamental role in the multidisciplinary management of patients with HNC over the last few decades. Nevertheless, some adverse events such as OM are a major concern for clinicians. OM is an acute inflammation that may initially manifest as redness of the mucous membrane, which eventually evolves into ulceration and formation of a pseudomembrane, leading to a temporary disruption of mucosal integrity [27]. OM pathogenesis, according to the most widely accepted biological model, appears to be a complex multistep process. This theory, proposed by Sonis et al. [28], describes an initiation phase, followed by upregulation and activation, signal amplification, ulceration, and ultimately a healing phase. It is crucial to understand that OM does not simply result from the epithelial injury caused during RT or CT; the epithelium, the underlying connective tissue, and the type of injury are the main characteristics of this mechanism, although other factors (the oral environment, immunological conditions, and performance status) also play a role in OM pathogenesis. Despite the wide interest in this topic, evidence regarding the risk factors for OM is limited, representing a challenge for healthcare professionals involved in the supportive care field [29]; in particular, mycetes harbouring in the oral cavity could play a pathogenic role in the onset and perpetration of OM. Few studies have investigated the possible role of *Candida* spp. in the onset of OM. 

The main objective of this study was to identify the clinical impact of oral *Candida* spp. colonisation on OM onset and its possible effects on OM severity. Although the prevalence of *Candida* spp. in the oral cavity was higher in patients with severe OM (20/36; 55.6%) than in other patients of the cohort (48/116; 41.4%), this correlation was not statistically significant (χ^2^ test, *p* = 0.097). Nevertheless, in our population, *Candida* spp. was identified as the only risk factor for EOM (OR: 5.13, 95% CI: 1.23–21.4 *p* = 0.04; Table 3). 

The early onset of severe OM is a critical issue in the HNC patients’ management, since it may further reduce the QoL of patients [30] and may increase the need for intensive supportive care and hospitalisation [31] and the need for treatment interruption, impairing the control of the disease [32]. Several studies have demonstrated a correlation between EOM and different CT regimens, RT dose per fraction, and treatment timing; however, [33] no studies have demonstrated a correlation between EOM and *Candida* spp. colonisation. Although the reason why *Candida* spp. colonisation resulted as the only risk factor for EOM is unclear, several hypotheses can be put forward. First, *Candida* spp. may accelerate the cascade of events leading to OM through their virulence factors, which can directly damage the epithelial layer by the production of cytolytic enzymes (proteinases, hemolysins, siderophores, and phospholipases) [34]. In addition, as highlighted by in vitro studies [35], *Candida* spp. blastospores may stimulate peripheral blood mononuclear cells to secrete tumour necrosis α (TNF-α) and the type 1 cytokine interferon-γ (IFN-γ), which contribute to OM pathogenesis [28]. However, it has been demonstrated that *Candida* spp. may influence bacterial growth, especially in the oral cavity, through different interaction mechanisms (adhesion and quorum sensing) [36]. This condition may cause a worsening of OM; in fact, it has been hypothesized that dynamic changes in the oral microbial community composition may be involved in OM pathogenesis [37]. HNC patients, often dentally compromised even before the initiation of RT [38], show microbial and inflammatory profiles different from healthy individuals [39], which probably contributes to the high incidence of OM in this group of patients.

Furthermore, an irradiated oral and oropharyngeal mucosa presents an ideal environment that favours opportunistic oral candidiasis; and ulcerations in the mucosal layer structure caused by OM may allow fungal penetration into the damaged tissue [20], enhancing the interactions between *Candida* spp. and the connective tissues. The ongoing inflammatory process, resulting in an anaerobic condition, could promote interactions between *Candida* spp. and pathogenic bacteria of the oral cavity [36]. Although not definitively proven, it is likely that RT may slightly change the oral microbiota composition, favouring the onset of oral candidiasis [40]. Another issue is that irradiated patients may often experience an impairment of the local and systemic immune systems, as highlighted by our findings (leukopenia was present in 12.5% of our samples) and previously published papers [41]. Furthermore, *Candida* spp. hyphae induce a higher production of IL-10, an immunosuppressive cytokine, indirectly worsening the local immune response [35]. Another factor that can be involved in this process is the reduction in salivary flow in irradiated patients. Although hyposalivation is considered a late-onset RT adverse event, it has been demonstrated that salivary flow may be reduced to 50–70% of the baseline after 10–16 Gy of RT [20,42]. Radiation induces a decrease in amylase activity, bicarbonate levels, and pH and a significant increase in the viscosity [43] of saliva, mainly due to the disruption of the mucin network [44], which could promote the transformation of *Candida* spp. into a pathogen. A recent observational study demonstrated that *Candida* spp. had higher biofilm formation capability in a population of irradiated patients than in healthy individuals [45]. Nevertheless, our findings do not demonstrate a direct correlation between hyposalivation and EOM; thus, further studies are required to evaluate this hypothesis.

Previous studies have investigated the role of *Candida* spp. in irradiated HNC patients, with contradictory results. Singh et al. [19] argued that OM is a risk factor for oral candidiasis, suggesting that *Candida* spp. may overinfect pre-existing lesions. Suryawanshi et al. [46] concluded that *Candida* spp. colonisation does not influence the severity of OM, whereas oropharyngeal candidiasis may play a role in increasing the duration and discomfort of OM. In contrast, Rao et al. [47] found that biweekly prophylactic administration of fluconazole, an antimycotic, during RT-CT for HNC reduced the incidence of OM. Although their study was retrospective and lacked a control group or a pre-RT *Candida* spp. colonisation assessment, our findings may confirm their interesting results. Further clinical trials, designed to investigate this specific outcome, are needed to confirm this hypothesis; performing an oral swab before starting RT should be considered a routine procedure during pre-RT dental evaluation, given its low cost and the potential impact of *Candida* spp. on OM. Future studies should evaluate the preventive eradication of *Candida* spp. and assess whether this therapy has a positive effect in reducing the incidence of EOM.

Other important results retrieved by this study include the incidence of OM in a cohort of patients with HNC and the role of other possible risk factors for OM. Among the studied population, while the overall incidence of OM was 78.9%, in accordance with the 80% estimate reported by Trotti et al. [13], only 36 of 152 (23.7%) patients developed severe OM, which is a lower rate than that reported in previous studies [30]. While severe OM occurred in 36 out of the 152 patents, no patient developed G4 or G5 OM. The low incidence of severe OM might have been due to the use of the VMAT technique, which helps reduce the duration of RT sessions and spares healthy tissues [48].

Based on the results of the multiple logistic regression, two main clinical parameters were significantly associated with a higher incidence of severe OM: the site of the tumour (OR: 1.29, 95% CI: 1–1.67, *p* = 0.05) and lymphocytopenia (OR: 8.2, 95% CI: 1.2–55.8, *p* = 0.03).

As expected, the larynx had the lowest incidence of severe OM (1/28 patients, 3.6%), followed by the hypopharynx (1/9 patients, 11%). In contrast, the nasopharynx was the site that showed the highest incidence (9/19 patients, 47.7%). This finding confirms the results of the vast majority of the literature [41,49], and can be due to the introduction of more targeted radiation therapy techniques and the use of increasingly smaller treatment volumes (PTV) in clinical practice, which helped to spare the oral and oropharyngeal mucosa, maintaining clinical efficacy on the tumoural tissues. 

Although a few studies have investigated the role of lymphocytopenia in the onset of OM [41] and found a correlation, the evidence is still limited [29]. It is reasonable to think that lymphocytopenia may have a direct or indirect role in modulating the inflammatory process leading to OM (i.e., dysregulation of the inflammatory processes and enhancement of the risk of bacterial colonisation of the epithelium, which stimulates cytokine production); however, the effective role of lymphocytes in the pathogenesis of OM needs to be clarified by other studies. Notably, Munneke et al., in a randomised clinical trial, highlighted how activated innate lymphoid cells are associated with reduced susceptibility to OM in a cohort of patients undergoing allogeneic haematopoietic stem cell transplantation (HSCT) [50], thus indirectly confirming our findings. 

Previous studies have revealed how concomitant CT can predict OM in patients with HNC [10,16,51]. Our results, in accordance with the previous results, showed a higher incidence (χ^2^ test, *p* < 0.05) of severe OM (29.1%) in RT-CT-treated patients than in patients treated with RT alone (16.7%). However, when included in a logistic regression model, this correlation was not statistically significant. Similarly, CT was not a statistically significant risk factor for EOM. It is likely that the number of patients treated with concomitant CT (86 patients) within our sample was not sufficient to draw statistically significant results.

This study has several strengths. Our results revealed how severe OM developed significantly earlier in patients with HNC with oral *Candida* spp. colonisation prior to initiation of RT. To our knowledge, this is the first study to prospectively investigate this relationship in patients with HNC undergoing RT, and its novelty lies in the fact that it highlights how oral *Candida* spp. can also play a crucial role in the onset of severe OM, apart from the simple overinfection of already existing lesions.

Furthermore, since this was a monocentric study, the RT treatment plan was always decided by the same radiotherapist (F.M.) with the same device, and the whole sample was homogeneous by mean total dose of radiation, which allowed the identification of risk factors different from the RT dose or RT technique.

This study also had some limitations. Its monocentric nature may have limited the reliability of our results on a larger scale, especially regarding the effects of different types of RT and other devices. Furthermore, because this study aimed to evaluate the onset of OM as a primary outcome, the duration of severe OM was not recorded. These data, however, need to be recorded in future studies and will contribute to a deeper description and understanding of the clinical impact of OM in terms of the number of visits during therapy, supportive therapy, economic burden, and perceived QoL changes. Another possible limitation of this study may be the lack of evaluation of the oral microbiota of included patients; it was not possible to carry out this analysis given its high economic impact. Nevertheless, given its significance in promoting the occurrence of oral candidiasis, further studies should evaluate its impact on opportunistic infection-related diseases.

## 5. Conclusions

In conclusion, our findings show that oral colonisation by *Candida* spp. is a predictive factor for EOM. Performing an oral swab test before initiation of RT should be considered as a routine procedure during pre-RT dental evaluation, given its low cost and the impact of *Candida* spp. on OM. Therefore, physicians, dentists, and otolaryngologists should be aware that OM prevention strategies should be implemented before the initiation of RT in patients with positive oral cavity swabs. Future studies should evaluate the preventive eradication of *Candida* spp. and assess whether this therapy has a positive effect in reducing the incidence of EOM.

## Figures and Tables

**Figure 1 cancers-14-04746-f001:**
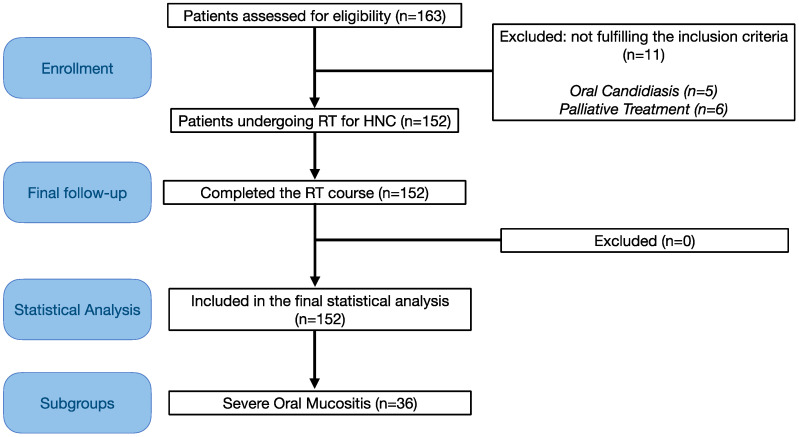
Strobe flow chart of the study.

**Table 1 cancers-14-04746-t001:** Basal patients’ characteristics, basal treatment characteristics of the studied population and treatment related toxicity parameters. SD: standard deviation; SCC: squamous cell carcinoma, RT: radiotherapy.

Variable	Group	N (%)
*Gender*	Men	103 (67.8%)
	Women	49 (32.2%)
	Total	152 (100%)

*Age*	Mean	60.3 (22–86; SD: 11.5)

*Comorbidities*	Yes	75 (49.3%)
	No	77 (50.7%)
	Total	152 (100%)

*Tumour Type*	SCC	130 (85.5%)
	Other types	22 (14.5%)
	Total	152 (100%)

*Tumour Stage*	Stage 1	10 (6.5%)
	Stage 2	27 (17.8%)
	Stage 3	38 (25%)
	Stage 4	77 (50.7%)
	Total	152 (100%)

*Tumour Site*	Hipopharynx	9 (5.9%)
	Larynx	28 (18.4%)
	Oral cavity	28 (18.4%)
	Oropharynx	38 (25%)
	Rhinopharynx	19 (12.6%)
	Salivary Glands	14 (9.2%)
	Other sites	16 (10.5%)
	Total	152 (100%)

*Smoking*	Smokers	87 (57.2%)
	Non smokers	65 (42.8%)
	Total	152 (100%)

*Surgery*	Performed	59 (38.8%)
	Not performed	93 (61.2%)
	Total	152 (100%)

*Chemotherapy*	Performed	86 (56.6%)
	Not Performed	66 (43.4%)
	Total	152 (100%)

*White Blood Cell Count*	Leucocytes (10^9^/L, Mean)	6.7 (1.6–15.1; SD: 2.5)
	Neutrophils (10^9^/L, Mean)	4.6 (0.7–11.8; SD: 2.1)
	Lymphocytes (10^9^/L, Mean)	2.2 (0.6–11.4; SD: 1.2)

*Leukopenia*	Yes	19
	No	133
	Total	152 (100%)

*Neutropenia*	Yes	4
	No	148
	Total	152 (100%)

*Lymphocytopenia*	Yes	15
	No	137
	Total	152 (100%)

*Salivary Flow*	mL (Mean)	2.6 (0–15; SD: 2.2)

*Hyposalivation (<2 mL)*	Yes (*n*. of Patients)	79 (51.9%)
	No (*n*. of Patients)	73 (48.1%)
	Total	152 (100%)

*Oral Candida*	Positive Oral Cavity Swab	68 (44.7%)
	Negative Oral Cavity Swab	84 (55.3%)
	Total	152 (100%)

*Total RT Dose*	Gy (Mean)	67.6 (50–72; SD: 3.9)

*Fractioning Schedule*	1.8 Gy/die	1 (0.7%)
	2 Gy/die	136 (89.5%)
	2.2 Gy/die	14 (9.2%)
	2.4 Gy/die	1 (0.7%)
	Total	152 (100%)

*Oral RT Dose*	Gy (Mean)	36.8 (0–75.2; SD: 16.9)

*Oropharynx RT Dose*	Gy (Mean)	50.3 (0–71.3; SD: 18.9)

*Oral Mucositis*	Absence	32 (21.1%)
	Grade 1	40 (26.3%)
	Grade 2	44 (28.9%)
	Grade 3	36 (23.7%)
	Grade 4	0 (0%)
	Total	152 (100%)

**Table 2 cancers-14-04746-t002:** Clinical variables of the studied population, according to the development of severe mucositis.

		Total Sample	Severe Mucositis		Statistical Significance
		*152 (100%)*	*Yes* *36* (23.7%)	*No* *116 (76.3%)*	

** *Gender* **	Male	103 (67.8%)	28 (27.2%)	75 (72.8%)	χ^2^ Test—*p* = 0.64
	Female	49 (32.2%)	8 (16.3%)	41 (83.7%)	
** *Age* **	Mean (Range; SD)	60.3 (22–86; 11.5)	58.7 (22–75; 10.5)	60.8 (29–86, 11.8)	ANOVA—*p* = 0.83
** *Total RT dose (Gy)* **	Mean (Range; SD)	67.6 (50–72; 3.9)	68.2 (60–70; 2.7)	67.4 (50–72; 4.2)	Mann–Whitney—*p* = 0.56
** *Mean oral cavity dose (Gy)* **	Mean (Range; SD)	36.8 (0–70; 16.9)	38.8 (0.8–65.9; 17.1)	36.2 (0–70; 16.9)	ANOVA—*p* = 0.427
** *Mean oropharynx dose (Gy)* **	Mean (Range; SD)	50.3 (0–71.3; 18.9)	50.7 (2.5–70.4; 22.3)	50.2 (0–71.3; 17.8)	Mann–Whitney—*p* = 0.22
** *Leucocytes (10^9^/l) ^a e^* **	Mean (Range; SD)	6.7 (1.6–15.1; 2.5)	4.9 (1.6–13.5; 2.4)	7.2 (3.0–15.1; 2.4)	**Mann–Whitney—*p* = 0.001**
** *Neutrophils (10^9^/l) ^b^* **	Mean (Range; SD)	4.6 (0.7–11.8; 2.1)	3.7 (0.7–11.8; 2.1)	4.8 (1.5–10.9; 2.0)	**Mann–Whitney—*p* = 0.01**
** *Lymphocytes (10^9^/l) ^c^* **	Mean (Range; SD)	2.2 (0.6–11.4; 1.2)	1.3 (0.7–5.8; 0.8)	2.5 (0.6–11.4; 1.1)	**Mann–Whitney—*p* = 0.001**
** *Leukopenia ^d^* **	Yes	19 (12.5%)	15 (41.7%)	4 (3.5%)	**χ^2^ Test—*p* = 0.001**
	No	133 (87.5%)	21 (58.3%)	112 (96.5%)	
** *Neutropenia ^e^* **	Yes	4 (2.6%)	3 (8.4%)	1 (0.9%)	**χ^2^ Test—*p* = 0.04**
	No	148 (97.4%)	33 (91.6%)	115 (99.1%)	
** *Lymphocytopenia ^f, i^* **	Yes	15 (9.9%)	13 (36.1%)	2 (1.8%)	**χ^2^ Test—*p* = 0.001**
	No	137 (90.1%)	23 (63.9%)	114 (98.2%)	
** *Comorbidities* **	Yes	75 (49.3%)	15 (20%)	60 (80%)	χ^2^ Test—*p* = 0.34
	No	77 (50.7%)	21 (27.3%)	56 (72.7%)	
** *Tumour type* **	SCC	130 (85.5%)	32 (24.6%)	98 (75.4%)	χ^2^ Test—*p* = 0.13
	Other types	22 (14.5%)	3 (13.6%)	19 (86.4%)	
** *Tumour site ^g, i^* **	Larynx	28 (18.4%)	1 (3.6%)	27 (96.4%)	**χ^2^ Test—*p* = 0.006**
	Oral cavity	28 (18.4%)	9 (32.1%)	19 (67.9%)	
	Oropharynx	38 (25%)	12 (31.6%)	26 (68.4%)	
	Rhinopharynx	19 (12.5%)	9 (47.4%)	10 (52.6%)	
	Salivary Glands	14 (9.2%)	1 (7.1%)	13 (92.9%)	
	Hypoparynx	9 (5.9%)	1 (11.1%)	8 (88.9%)	
	Other sites	16 (10.5%)	3 (18.7%)	13 (81.3%)	
** *Tumour stage* **	Stage I	10 (6.6%)	2 (20%)	8 (80%)	χ^2^ Test—*p* = 0.69
	Stage II	27 (17.8%)	4 (14.8%)	23 (85.2%)	
	Stage III	38 (25%)	10 (26.3%)	28 (73.7%)	
	Stage IV	77 (50.6%)	20 (25.9%)	57 (74.1%)	
** *Chemotherapy ^h^* **	Yes	86 (56.6%)	25 (29.1%)	61 (70.9%)	**χ^2^ Test—*p* = 0.05**
	No	66 (43.4%)	11 (16.7%)	55 (83.3%)	
** *Surgery* **	Performed	59 (38.8%)	11 (18.6%)	48 (81.4%)	χ^2^ Test—*p* = 0.32
	Non performed	93 (61.2%)	25 (26.9%)	68 (73.1%)	
** *Smoking* **	Yes	87 (57.2%)	21 (24.1%)	66 (75.9%)	χ^2^ Test—*p* = 0.52
	No	65 (42.8%)	15 (23.1%)	50 (76.9%)	
** *Oral candida swab* **	Positive	68 (44.7%)	20 (29.4%)	48 (70.6%)	χ^2^ Test—*p* = 0.097
	Negative	84 (55.3%)	16 (19%)	68 (81%)	
** *Hyposalivation (<2 mL)* **	Yes	79 (51.9%)	60 (75.9%)	19 (24.1%)	χ^2^ Test—*p* = 0.53
	No	73 (48.1%)	56 (76.7%)	17 (23.3%)	

^a^ Correlation between leucocytes and severe OM—Mann–Whitney test—*p* < 0.05. ^b^ Correlation between neutrophils and severe OM—Mann–Whitney test—*p* < 0.05. ^c^ Correlation between lymphocytes and severe OM—Mann–Whitney test—*p* < 0.05. ^d^ Correlation between Leukopenia and severe OM—χ^2^ Test—*p* < 0.05. ^e^ Correlation between neutropenia and severe OM—χ^2^ Test—*p* < 0.05. ^f^ Correlation between lymphocytopenia and severe OM—χ^2^ Test—*p* < 0.05. ^g^ Correlation between tumour site and severe OM—χ^2^ Test—*p* < 0.05. ^h^ Correlation between chemotherapy and severe OM—χ^2^ Test—*p* = 0.05. ^i^ Multiple Logistic Regression showed how tumour site and lymphocytopenia are risk factors for severe OM: tumour site—OR: 1.29, 95% CI: 1–1.67, *p* = 0.05. Lymphocytopenia-OR: 8.2, 95% CI: 1.2–55.8, *p* = 0.03.

**Table 3 cancers-14-04746-t003:** Clinical variables of the studied population, according to the development of severe mucositis. WBC: white blood cell count.

		Total Sample	Early Onset Severe OM		Statistical Significance
		*36 (100%)*	*Yes* *19 (52.8%)*	*No* *17 (47.2%)*	

** *Gender* **	Male	28 (77.8%)	13 (46.4%)	15 (53.6%)	χ^2^ Test—*p* = 0.24
	Female	8 (22.2%)	6 (75%)	2 (25%)	
** *Comorbidities* **	Yes	15 (43.1%)	8 (53.3%)	7 (46.7%)	χ^2^ Test—*p* = 0.9
	No	21 (56.9%)	11 (52.4%)	10 (47.5%)	
** *Age* **	Mean (Range; SD)	58.7 (22–75; 10.5)	59.1 (22–70, 11.7)	58.3 (43–75; 9.2)	ANOVA—*p* = 0.19
** *Total rt dose (Gy)* **	Mean (Range; SD)	68.2 (60–70; 2.7)	68.4 (60–70; 2.7)	68 (60–70; 2.8)	Mann-Whitney—*p* = 0.69
** *Mean oral cavity dose (Gy)* **	Mean (Range; SD)	38.8 (0.8–65.9; 17.1)	37.9 (3.4–65.9; 17.1)	39.7 (0.8–63.7; 17.6)	ANOVA—*p* = 0.75
** *Mean oropharynx dose (Gy)* **	Mean (Range; SD)	50.7 (2.5–70.4; 22.3)	49.4 (2.7–70.4; 24.6)	52 (2.5–69.2; 19.9)	Mann-Whitney—*p* = 0.75
** *WBC (10^9^/l)* **	Leucocytes (Range; SD)	4.9 (1.6–13.5; 2.4)	5.1 (2.5–8.9; 1.9)	4.6 (1.6–13.5; 2.9)	Mann-Whitney—0.45
	Neutrophils (Range; SD)	3.7 (0.7–11.8; 2.1)	3.7 (1.7–7.1; 1.5)	3.5 (0.7–11.7; 2.4)	Mann-Whitney—0.21
	Lymphocytes (Range; SD)	1.3 (0.7–5.8; 0.8)	1.4 (0.7–5.7; 1.1)	1.1 (0.6–1.7; 0.3)	Mann-Whitney—0.18
** *Leukopenia* **	Yes	15 (41.7%)	7 (46.7%)	8 (53.3%)	χ^2^ Test—*p* = 0.73
	No	21 (58.3%)	12 (57.1%)	9 (42.9%)	
** *Neutropenia* **	Yes	3 (8.4%)	0 (-)	3 (100%)	χ^2^ Test—*p* = 0.09
	No	33 (91.6%)	19 (57.6%)	14 (42.4%)	
** *Lymphocytopenia* **	Yes	13 (36.1%)	5 (38.4%)	8 (61.6%)	χ^2^ Test—*p* = 0.3
	No	23 (63.9%)	14 (60.9%)	9 (39.1%)	
** *Tumour type* **	SCC	32 (88.9%)	17 (53.1%)	15 (46.9%)	χ^2^ Test—*p* = 0.79
	Other types	4 (11.1%)	2 (50%)	2 (50%)	
** *Tumour site* **	Larynx	1 (2.8%)	1 (100%)	0 (-)	χ^2^ Test—*p* = 0.17
	Oral cavity	9 (25%)	5 (55.6%)	4 (44.4%)	
	Oropharynx	12 (66.7%)	4 (33.3%)	8 (66.7%)	
	Rhinopharynx	9 (25%)	6 (66.7%)	3 (33.3%)	
	Salivary Glands	1 (2.8%)	0 (-)	1 (100%)	
	Hypoparynx	1 (2.8%)	0 (-)	1 (100%)	
	Other sites	3 (2.8%)	3 (100%)	0 (-)	
** *Tumour stage* **	Stage I	2 (5.6%)	2 (100%)	0 (-)	χ^2^ Test—*p* = 0.43
	Stage II	4 (11.2%)	1 (25%)	3 (75%)	
	Stage III	10 (27.8%)	6 (60%)	4 (40%)	
	Stage IV	20 (55.6%)	10 (50%)	10 (50%)	
** *Chemotherapy* **	Yes	25 (69.4%)	14 (56%)	11 (44%)	χ^2^ Test—*p* = 0.72
	No	11 (30.6%)	5 (45.5%)	6 (54.5%)	
** *Surgery* **	Performed	11 (30.6%)	5 (45.5%)	6 (54.5%)	χ^2^ Test—*p* = 0.72
	Non performed	25 (69.4%)	14 (56%)	11 (44%)	
** *Smoking* **	Yes	21 (58.3%)	11 (52.4%)	10 (47.6%)	χ^2^ Test—*p* = 0.61
	No	15 (41.7%)	8 (53.3%)	7 (46.7%)	
** *Oral candida swab ^a^* **	Positive	20 (55.6%)	14 (70%)	6 (30%)	**χ^2^ Test—*p* = 0.04**
	Negative	16 (44.4%)	5 (31.3%)	11 (68.7%)	
** *Hyposalivation (<2 mL* ** ** *)* **	Yes	17 (47.2%)	5 (29.4%)	12 (70.6%)	χ^2^ Test—*p* = 0.06
	No	19 (52.8%)	12 (63.2%)	7 (36.8%)	

^a^ Correlation between oral candida and early onset severe OM—χ^2^ Test—*p* < 0.05; (OR: 5.13, 95% CI: 1.23–21.4 *p* = 0.04).

## Data Availability

The data presented in this study are available on request from the corresponding authors. The data are not publicly available because of privacy concerns.

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
