# Peer review of "Oral Candida spp. Colonisation Is a Risk Factor for Severe Oral Mucositis in Patients Undergoing Radiotherapy for Head & Neck Cancer: Results from a Multidisciplinary Mono-Institutional Prospective Observational Study"

_cancers, 2022, doi:10.3390/cancers14194746_

Round 1
Reviewer 1 Report
I think it is necessary to present the material and methods in the form of topics, to facilitate the understanding of the study carried out. Some tables are too large and need to be rewritten.

Author Response
Dear Editor,
thank you for your e-mail. Attached please find a copy of the revised paper according to the helpful suggestions of the reviewers. Changes are highlighted in the text according to the Microsoft Word track changes mode. Below, you can find a point-to-point rebuttal to the reviewers’ comments.
Looking forward to hearing from you, please accept my best regards.
Reply to reviewer 1
I think it is necessary to present the material and methods in the form of topics, to facilitate the understanding of the study carried out. Some tables are too large and need to be rewritten.
We would like to thank the reviewer for his opinion. We modified the materials and methods sections and the tables according to his useful suggestions

Reviewer 2 Report
The article by Rupe et al. summarizes the results of a clinical study on the influence of Candida spp. colonization on the development of severe oral mucositis in head and neck cancer patients treated with radiation therapy. Cwell andida colonization and lymphocytopenia were risk factors for early onset severe oral mucositis in these patients.
This study is of high interest to clinicians treating HNC patients, especially radiooncologists. The manuscript is well written and study design, methods and results overall well described and discussed.
A minor point that should be addressed before acceptance is a description of the reasons why 11 patients were excluded to address possible biases of the results.
Reviewer 3 Report
In abstract is stated … oral mucositis ( grade 3/4) but further in text is stated grade III-IV), please unify this.The reference numbers should be placed in square brackets according to MS preparation guidelines [1,2] not [1] [2]….( see interduction and materials and metod section). My main concern is due to methodology.The timeline of OM assessment is not clear. Also, it is not clear if t patients did receive any OM therapy during RT treatment.
